# Question Answer System: A State-of-Art Representation of Quantitative and Qualitative Analysis

Bhushan Zope [1,*], Sashikala Mishra [1], Kailash Shaw [1], Deepali Rahul Vora [1], Ketan Kotecha [2] and Ranjeet Vasant Bidwe [1]

1   Symbiosis Institute of Technology, Symbiosis International (Deemed University) (SIU), Lavale, Pune 412115, India

2   Symbiosis Centre for Applied Artificial Intelligence (SCAAI), Symbiosis Institute of Technology, Symbiosis International (Deemed University) (SIU), Lavale, Pune 412115, India

*   Correspondence: bhushan.zope.phd2021@sitpune.edu.in

**Abstract:** Question Answer System (QAS) automatically answers the question asked in natural language. Due to the varying dimensions and approaches that are available, QAS has a very diverse solution space, and a proper bibliometric study is required to paint the entire domain space. This work presents a bibliometric and literature analysis of QAS. Scopus and Web of Science are two well-known research databases used for the study. A systematic analytical study comprising performance analysis and science mapping is performed. Recent research trends, seminal work, and influential authors are identified in performance analysis using statistical tools on research constituents. On the other hand, science mapping is performed using network analysis on a citation and co-citation network graph. Through this analysis, the domain's conceptual evolution and intellectual structure are shown. We have divided the literature into four important architecture types and have provided the literature analysis of Knowledge Base (KB)-based and GNN-based approaches for QAS.

**Keywords:** question answering system; bibliometric analysis; natural language processing; machine comprehension

## 1. Introduction

Question answer system (QAS) is standard Natural Language Processing (NLP) task. In this digital era, we are drowned in a sea of information. We have web search engines that help us sail through the information, but their application is limited and could not help us beyond certain limits. While looking for the answers, web search engines can only direct to the answer's probable locations, but one must sort through to find the answer. It is fascinating to have an automatic system that can fetch/generate the answer from retrieved documents instead of only displaying them to the user. Thus, QAS finds the natural language answers for the natural language questions.

Since QA is an intersection of NLP, Information Retrieval (IR), Logical Reasoning, Knowledge Representation, Machine learning, semantic search, QA can be used to quantifiably measure any Artificial Intelligence (AI) system's understanding and reasoning capability [1,2]. Efforts have been taken in this respect since the 1960s. However, due to the availability of excellent computation power and emergence of many state-of-the-art deep learning algorithms, this field has gained momentum lately. These latest deep learning models have performed better than humans on single paragraph question answering benchmarks, including SQuAD [3–5]. However, a QAS system capable of answering complex questions is still elusive [6].

All QA systems can be classified based on the type of questions it is trying to answer. Different types of questions are given in Figure 1. Every type comes with its own set of requirements and needs different treatment. However, we can broadly divide the QAS process into three parts, as shown in Figure 2. Part one deals with understanding

the question asked in natural language, while part two is about finding the appropriate background information needed to answer the question. Part three is about finding the probable answers and selecting the most appropriate from them. Many techniques have been used to cater to each phase. To match the reading comprehension level of humans, Ding et al. [7] explains three main challenges of the QA system: 1. Reasoning ability, 2. Explainability, and 3. Scalability.

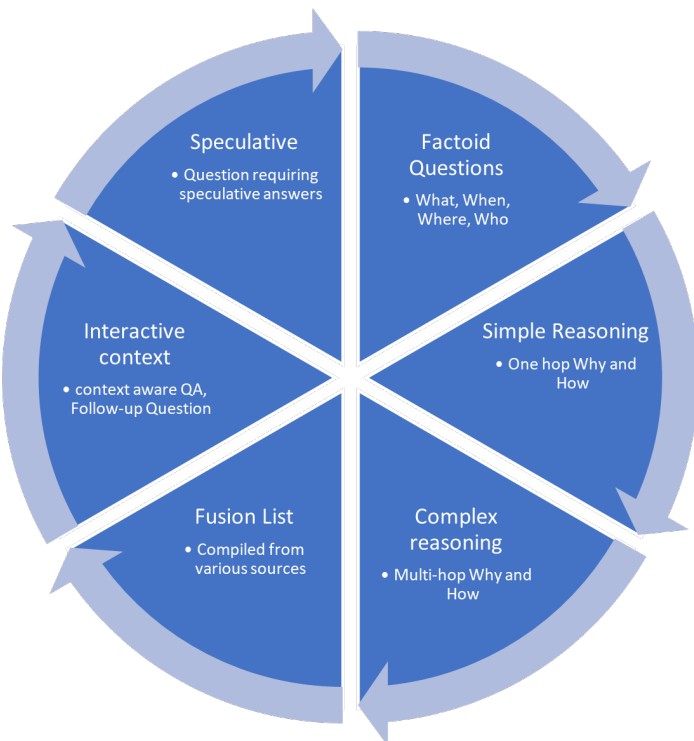

**Figure 1.** Types of Questions.

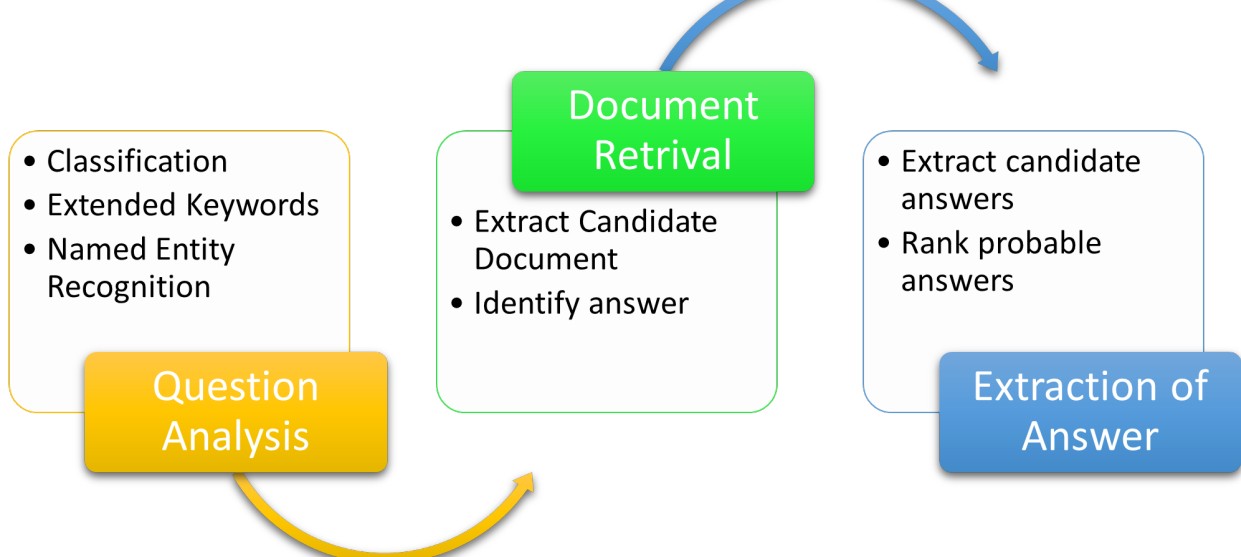

**Figure 2.** Phases of QAS.

Bibliometric analysis has a history of 60–70 years, but a remarkable increase in popularity has been seen in the last couple of decades. Bibliometric methodology analyses the literature from a quantitative perspective [8]. Thus, quantitative techniques are applied to

the bibliometric data like citation, publication, and keywords. Such bibliometric data is rigorously analyzed to uncover the interesting trends and relationships between various research constituents. Due to databases like SCOPUS or Web of Science, collecting a large amount of data has become easier. Also, many software tools like VOSViewer, Gephi, and bibexcel have been created to transform, analyze, and visualize such massive data. This has further led to recently increasing the popularity of bibliometric analysis.

Even though the Question Answering domain has been studied since the 1960s, it is a dynamic and still relevant field. It amalgamates various important fields like IR, NLP, Knowledge representation, logical reasoning, Machine learning, and many more. Due to this rich context, many diverse approaches have been proposed. Many qualitative survey papers describe the publications and their methodologies used; but due to a vast number of publications, systematic literature review becomes infeasible. Also it will be very interesting to study the effect of interaction between these important fields and the solutions emerged from it. Only a single previous publication byBlooma et al. [9] attempts to perform the bibliometric analysis on the QA domain. However, the time span considered for that study was very small (2000–2007), and only co-word analysis was done. Hence, a thorough bibliometric analysis of bibliometric data of the QA domain is still not done. To bridge this research gap, we have explored the following research questions in this study.

1.　To uncover the research themes and their evolution in the QA domain
2.　To identify important constituents and their contribution to the domain
3.　To identify seminal works/ publications in the field of QA

This paper is divided into five sections. To collect the relevant data for the analysis, enormous databases are need to be searched. Hence searching strategy is essential, which is described in Section 2. Section 3 discusses the results of the quantitative analysis with graphical visualizations. Section 4 contains the qualitative analysis, followed by Section 5, concluding the article with important findings.In bibliometric analysis, a vast amount of documents are considered and analyzed. Hence, the methodology framework plays a crucial role in the success of the study. Figure 3 shows the framework methodology of this study. Authors of this paper have designed the framework as per bibliometric analysis technique toolbox explained by Donthu et al. [10].

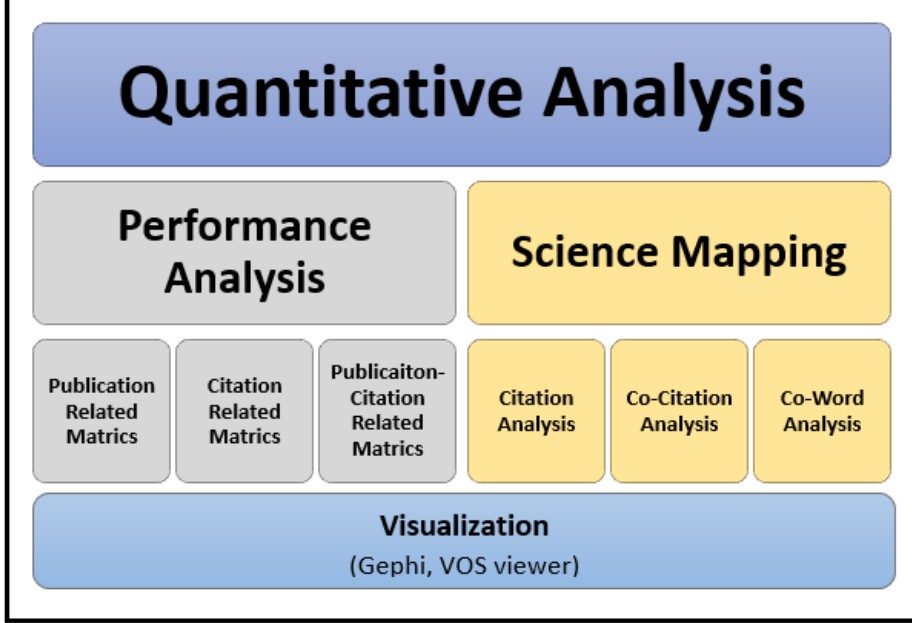

**Figure 3.** Methodological Framework.

There are basically two dimensions to analyse any scientific field. First dimension try to find out the contribution of a particular research constituent to a given field. Such kind of analysis is called as performance analysis. Contribution can be represented using various metrics such as Year-wise publication, Most productive countries, Most cited publications, Top publishers etc. another dimension try to find the inter-relationships between these research constituents. hence research constituent like authors, publications, keywords are represented as interconnected network and network analysis is performed to identify the key nodes in network. Such analysis is called as science mapping. Our results of performance analysis and science mapping are explained in Sections 3.1 and 3.2.

## 2. Search Strategy

Bibliometric Analysis is carried out on the published documents to explore the "question answering system" research space. As per Figure 4, subsequent search strategy is followed. The two most important bibliometric databases (viz. Web of Science (WOS) and SCOPUS) are queried to find out the related documents. The term "Question-Answer" is very general and used in many subject areas like medicine, psychology, social science, art & humanities, etc. After careful examination of all the subject areas, it is observed that results from subject areas except Computer Science and Engineering are irrelevant to this study. Also, while verifying some sample documents manually, it was observed that few articles talked about the question-answering systems based on images or visuals or multimedia. However, our scope is strictly related to a text-based QA system. Hence, such documents are also excluded from this study. Section 2.1 briefly discusses the procedure of data collection and Section 2.2 describes the data pre-processing methods.

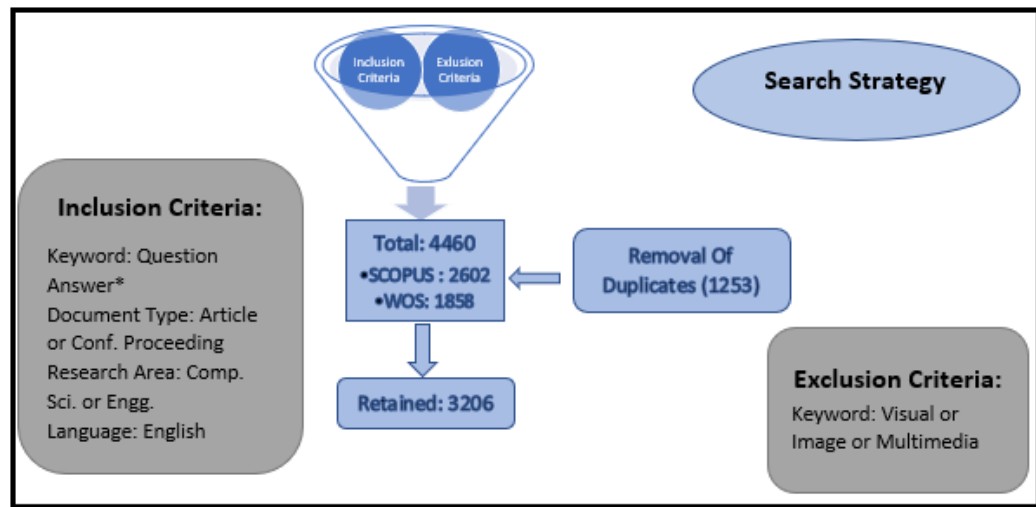

**Figure 4.** Search Strategy. * is wild card used to represent any or no group of character.

### 2.1. Sources and Methods

Science Citation Index Expanded (SCI-EXPANDED) and Conference Proceedings Citation Index—Science (CPCI-S) edition of WOS core collection and SCOPUS database are used to collect the documents. Queries given in Table 1 are executed on respective databases on Dec-2021. Due to the problems discussed earlier, results are further refined by following inclusion criteria.

1.   Document Type: Proceeding Paper or Article
2.   Research area: Computer science or engineering
3.   Language: English

**Table 1.** Queries.

| Database | Query | No. of Documents |
|----------|-------|------------------|
| WOS | "question answer *" (Title) not Visual (Title) not image (title) not Multimedia (title) | 1858 |
| Scopus | TITLE ("question Answer *" and not (visual or image or video)) AND (LIMIT-TO (SRC-TYPE,"p") OR LIMIT-TO (SRCTYPE,"j")) AND (LIMIT-TO (SUBJAREA,"COMP") OR LIMIT-TO (SUBJAREA,"ENGI")) AND (LIMIT-TO (DOCTYPE,"cp") OR LIMIT-TO (DOCTYPE,"ar")) AND (LIMIT-TO (LANGUAGE,"English")) | 2601 |

"*" is wild card used to represent any or no group of character.

### 2.2. Data Pre-Processing

Labels used to denote the same bibliometric information are different in both databases. The type of information also varies. Due to this inconsistent representation, we cannot directly combine the results from both databases. The maximum-common-attribute (MCA) set of both sources is identified to join the data, and data are combined manually in a field-wise manner. This 'merged-dataset' containing (4459 documents) is then used for most of the analysis. For analysis that requires the attribute not present in MCA, experimentation is done separately on WOS and SCOPUS data.

Furthermore, there are many inconsistencies in the form of misspelling and typing errors in title or journal names and non-standardized ways of representing the date. Due to these inconsistencies, pre-processing is necessary for this data. N-gram fingerprint key collision algorithm and a nearest neighbor algorithm is applied to cluster similar document titles. We have used the openRefine software tool for the same. Later after manual inspection of each cluster, all the titles in the same cluster are replaced by one consistent representation.

Both the databases can index a same document, and also our data may contain some duplicate entries. 1253 duplicate documents were identified and removed using MS-EXCEL software from the merged dataset, and final unique 3206 documents were retained. These docuements are then used for quantitative survey explained in Section 3.

### 3. Quantitative Survey

There are two vital points in a quantitative survey: (1) the contribution of individual research constituents and (2) the relationship between those research constituents. While performance analysis, discussed in Section 3.1 highlights the contribution of research constituents, in Section 3.2, science Mapping discusses their relationship.

### 3.1. Performance Analysis

Performance analysis is a study to discover the contribution of various research constituents to the given field. Typically, this analysis is more on the descriptive side. Despite the descriptive nature, it helps to recognize the importance of the various constituents. There are various metrics available to perform the performance analysis. Some metrics are related to a number of publications, which signifies the productivity of the field, while few metrics are citation-based, which measures the impact or significance. Few other metrics like total-publications-with-citation and h-index involve publication and citation. All three kinds of measures are considered in this study. Section 3.1.1 discusses the publication-related measures, while Section 3.1.2 deals with citation-based measures. Afterward, publication-citation-based measures are discussed in Section 3.1.3.

#### 3.1.1. Publication Related Metrics

The existence of the research paper in the early 1960s, as seen in Figure 5, indicates that automatically answering a question or reading comprehension was a significant area in the 1960s. However, this field gained significant traction in the 2000s decade. Besides, the availability of massive computing resources and the emergence of state-of-the-art (SOTA) language models in the 2010s and 2020s fuelled the research in this area, which is also

clearly visible in the figure. The fact that 50% of the research publications have been produced in the last six years demonstrates that researchers are pursuing this field more rigorously now.

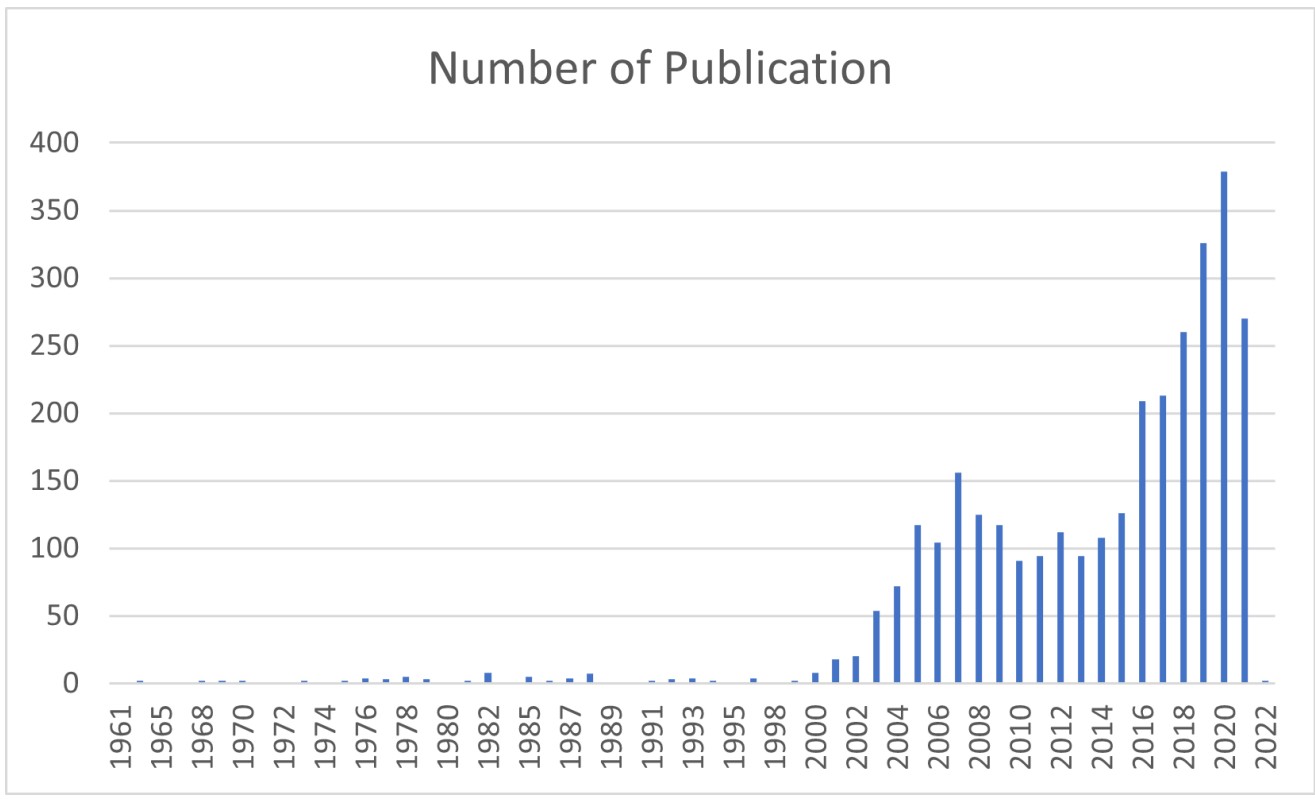

**Figure 5.** Year-wise publication.

Figure 6 represents the publication count by geographical area. This kind of analysis is required to understand the opportunities available worldwide in the field. China and USA are the top two countries producing the majority of the research papers while India and Germany come distant third and fourth in the ranking. China with 520, USA with 368, India with 140, and Germany with 119 publications; these top 4 countries have produced more than 50% of publications. Table 2 shows the exact publication count for the top 10 countries.

**Table 2.** Most productive countries.

| Country | Number of Publications |
|---|---|
| China | 520 |
| United States | 368 |
| India | 140 |
| Germany | 119 |
| Spain | 97 |
| Japan | 90 |
| England | 58 |
| South Korea | 55 |
| Canada | 51 |
| Italy | 51 |

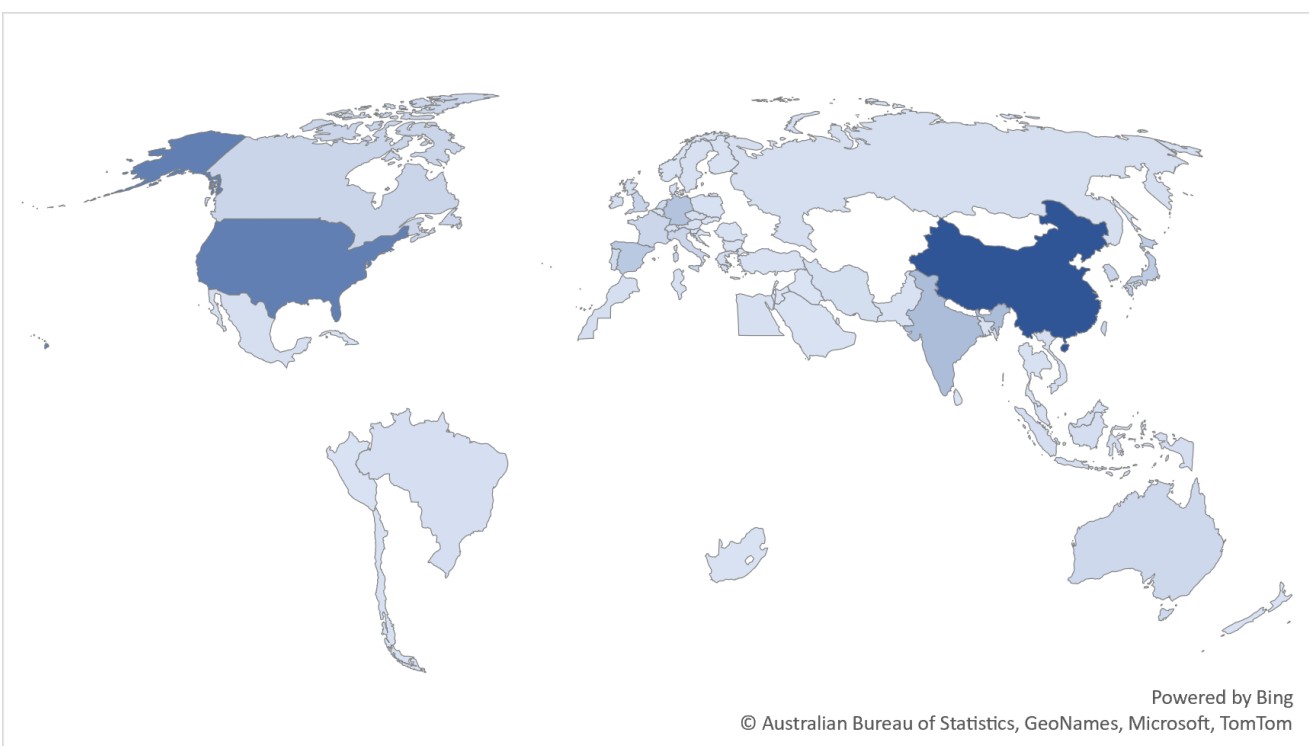

**Figure 6.** Most productive Countries.

Figure 7 shows the top 15 authors from WOS and SCOPUS datasets. Penas A has the highest number of publications; most are indexed in both databases. Few other influential authors are Nakov P, Moschitti A, and Lehman J.

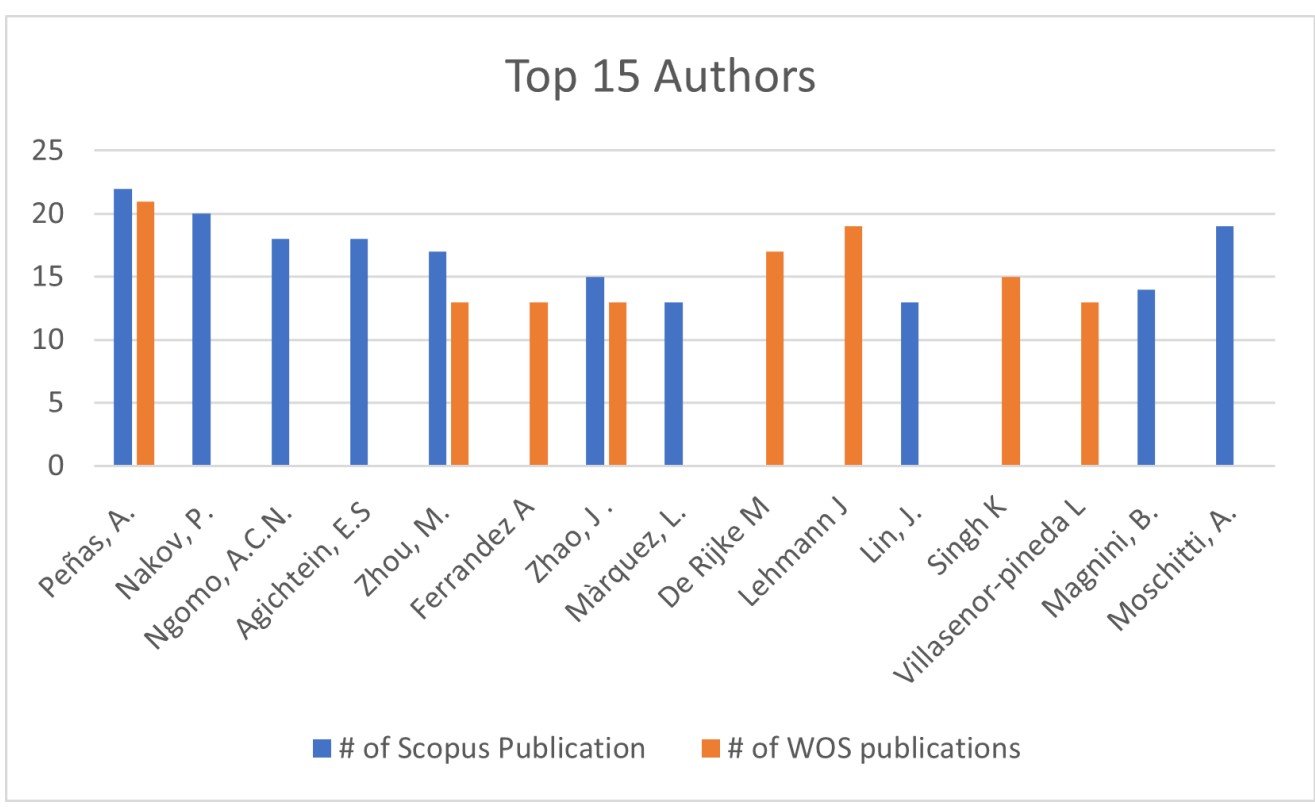

**Figure 7.** Top 15 authors.

Figures 8 and 9 point to the top sources publishing the research indexed in SCOUS and WOS, respectively, in the question-answer domain. The majority of the research documents indexed in WOS are published in Spring nature, IEEE, ACM, Elsevier, while ceur workshop proceedings, NIST Special Publication, ACM conferences, IEEE access are most preferred sources in case of SCOPUS.

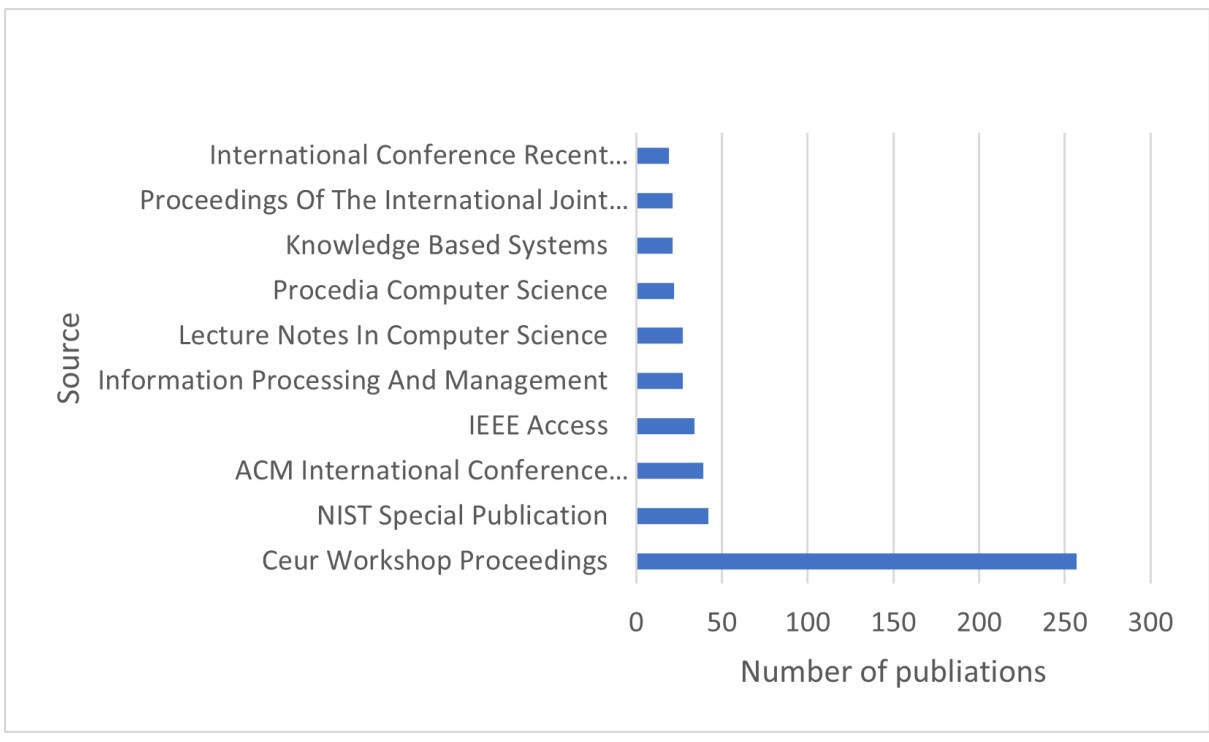

**Figure 8.** Top Publishers (SCOPUS).

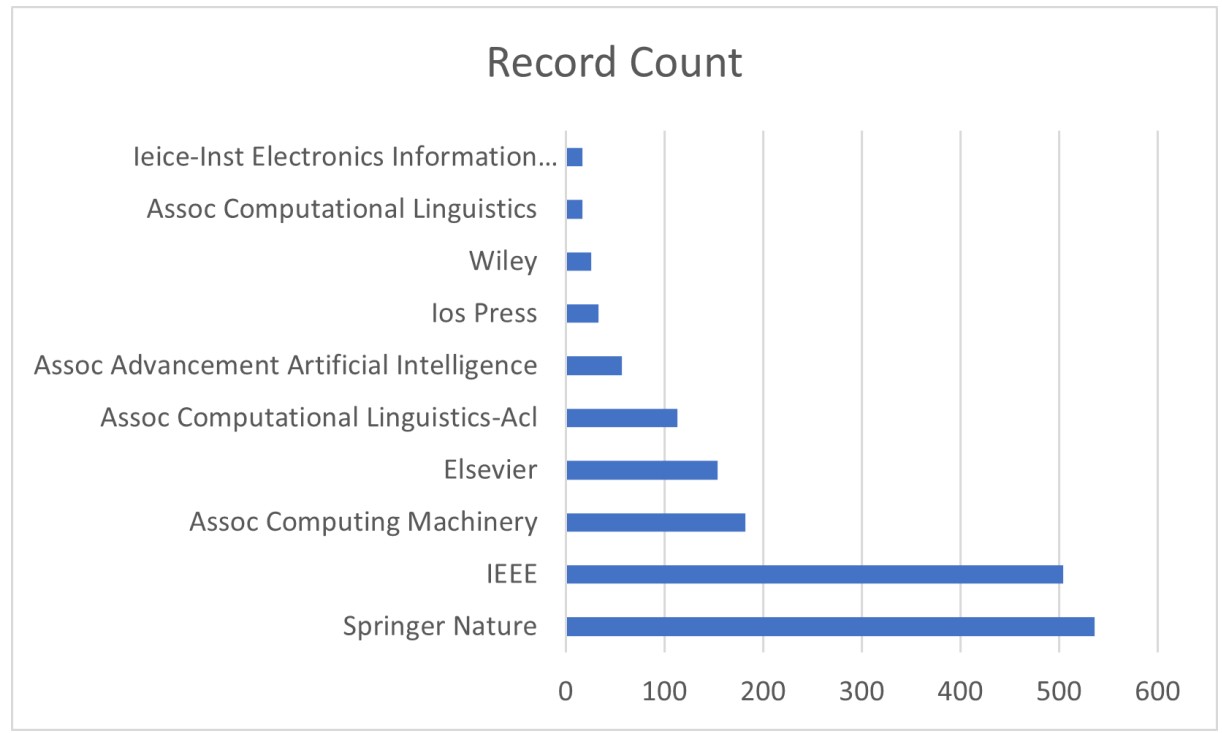

**Figure 9.** Top Publishers (WOS).

### 3.1.2. Citation-Related Metrics

The number of citations and trends in the citation indicates the research field's relevance. Extracted documents have a total of 32182 citations and 17.11 citations per publication. Such a high number of total publications and massive average citations per publication ratio indicates that this field is still relevant, and many researchers are currently working in this domain. Table 3 and 4 list the top 10 highly cited publications from SCOPUS and WOS in the last five years. Out of 15 unique publications listed in Table 3 and 4 together, six publications are directly related to either the 'knowledge graph' or 'knowledge base' keyword, highlighting that these are the latest research trends in the QA domain.

**Table 3.** Highly cited publications in the last five years (SCOPUS).

| Sr. No. | Reference | 2017 | 2018 | 2019 | 2020 | 2021 | Total |
|---------|-----------|------|------|------|------|------|-------|
| 1 | Wang et al. [11] | 3 | 63 | 94 | 119 | 43 | 322 |
| 2 | Lukovnikov et al. [12] | 5 | 18 | 32 | 45 | 41 | 143 |
| 3 | Hao [13] | 0 | 14 | 31 | 55 | 30 | 130 |
| 4 | Yang [14] | 0 | 0 | 11 | 78 | 32 | 122 |
| 5 | Xiong et al. [15] | 7 | 23 | 25 | 47 | 11 | 115 |
| 6 | Yu et al. [16] | 0 | 8 | 27 | 43 | 30 | 108 |
| 7 | Huang et al. [17] | 0 | 0 | 8 | 36 | 62 | 107 |
| 8 | Abujabal et al. [18] | 1 | 15 | 29 | 31 | 25 | 101 |
| 9 | Wang [19] | 0 | 4 | 21 | 45 | 25 | 95 |
| 10 | Khot et al. [20] | 0 | 7 | 24 | 45 | 19 | 95 |

**Table 4.** Highly cited publications in the last five years (WOS).

| Sr. No. | Reference | 2017 | 2018 | 2019 | 2020 | 2021 | Total |
|---------|-----------|------|------|------|------|------|-------|
| 1 | Wang et al. [11] | 1 | 34 | 65 | 52 | 30 | 182 |
| 2 | Lukovnikov et al. [12] | 3 | 13 | 22 | 24 | 18 | 80 |
| 3 | Das et al. [21] | 0 | 2 | 22 | 25 | 20 | 69 |
| 4 | Hao [13] | 0 | 7 | 21 | 28 | 12 | 68 |
| 5 | Cui et al. [22] | 0 | 6 | 17 | 20 | 14 | 57 |
| 6 | Yu et al. [16] | 0 | 5 | 19 | 13 | 15 | 53 |
| 7 | Hoeffner et al. [23] | 0 | 10 | 16 | 11 | 12 | 49 |
| 8 | Neshati et al. [24] | 0 | 2 | 15 | 19 | 10 | 46 |
| 9 | Abujabal et al. [18] | 0 | 6 | 17 | 12 | 7 | 42 |
| 10 | Esposito et al. [25] | 0 | 0 | 0 | 16 | 22 | 38 |

### 3.1.3. Publication-Citation-Related Metrics

Even though the total number of publications in any given field is sufficient to show the activeness of a particular field, the research's impact can also be measured using the h-index. H-index is a quantitative metric that indicates that there are n publications with at least n citations. Hence higher the h-index, higher the importance and significance. As seen in Figure 5, most of the publications are from recent years, and hence due to their citation time window, they haven't received any citations yet. However, out of 3206 documents under consideration, 1881 documents have at least one citation, and the h-index of the collection is 80.

*3.2. Science Mapping*

Science mapping helps us to study the internal structure of research constituents. Its primary usage is to understand the relationships between various entities. In science mapping, how the research constituents like citation, authors, documents, and keywords are connected is presented spatially. The interaction between these entities brings out this field's structure and highlights how this field has been explored. Out of many approaches to science mapping, this study uses citation analysis, Co-citation analysis, and Co-word analysis.

The interaction between these constituents is often viewed in some kind of graph or a network. Many network metrics are deployed to enrich the assessment of bibliometric information. Network metrics like degree of centrality, betweenness centrality, eigenvector centrality, and page rank can be used to understand the network more clearly. The role of the entity in the overall network and its relative importance can be assessed using these network metrics.

- Degree of centrality: Most basic metric of all is a degree of centrality, which measures the connection of an entity (which can be a document, author, keyword) to other entities in a network. So, if a particular entity is connected to many other entities, that entity has a high degree of centrality. Thus, representing its overall importance in the network.
- Betweenness centrality: It measures the ability of an entity (node) to transmit the information of one part of a network to other parts of the network. So, an entity with a high degree of betweenness centrality plays a crucial role in connecting, otherwise disconnected, part of a network. Such entities more often act as bridges between sub-fields in a research area.
- PageRank: PageRank is the most widely used method to find the importance of an entity. An entity with a small degree of centrality can profoundly affect the network if it influences highly connected entities. This fact is considered in PageRank.
- Eigenvector Centrality: eigenvector centrality is another metric to measure the influence on the overall network. The position of a node determines its overall influence on a network. Thus, the node connected to many highly connected nodes must have influenced the good part of a network; this forms the basic idea behind Eigenvector centrality. Thus, eigenvector centrality is high if a node is connected to many highly connected nodes.

Clustering is also an essential tool that can enrich the bibliometric assessment. Clustering can be used to identify the various sub-themes. By observing the clusters created, one can visualize the major underlying themes and how they have evolved over the period. In this study, VOSViewer [26] and Gephi [27] have been used for visualization, network analysis, and clustering. Microsoft Excel is used for basic visualization like chart plotting. VOSViewer is a popular visualization tool for bibliometric data. Gephi, on the other hand, is a tool popularly used for network analysis. It can also be used for visualization. In this study, networks are constructed using VOSViewer, and Gephi has been used to perform the network analysis.

### 3.2.1. Citation Analysis

Citation analysis is the most basic analysis for science mapping. Publications are linked based on their citation linkage, i.e., if one publication cites other, there is a link from the first publication to the second. This way network is formed. Thus, the citation becomes the basis for an intellectual relationship between publications.

Citation analysis is done on both databases separately. Table 5 discusses the strategy for this analysis. 1009 publications out of 2601 from SCOPUS and 494 publications out of 1858 from WOS fulfill the minimum criteria of 5 citations. These publications are used for network formation. The greatest connected component is further considered for the network analysis. Visualization is then created using Gephi's Fruchterman Reingold

layout. Looking at similarity in citation, nodes are clustered and represented using cluster color in the visualization. Figure 10 and 11 shows the citation network for SCOPUS and WOS respectively.

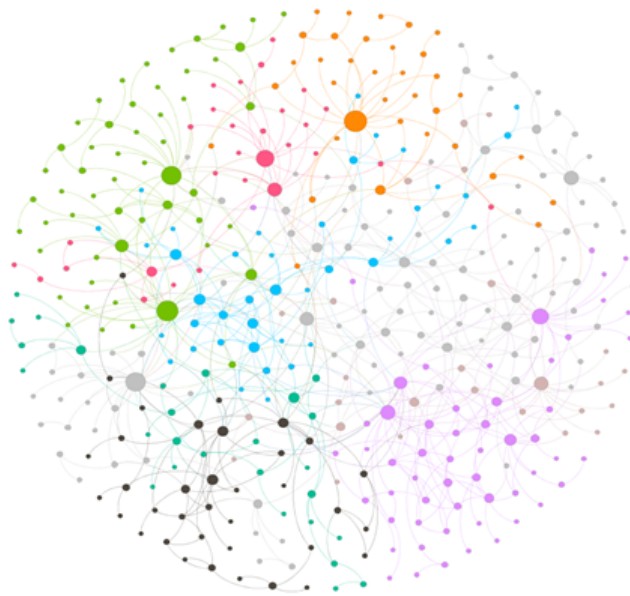

**Figure 10.** Citation Network (SCOPUS)

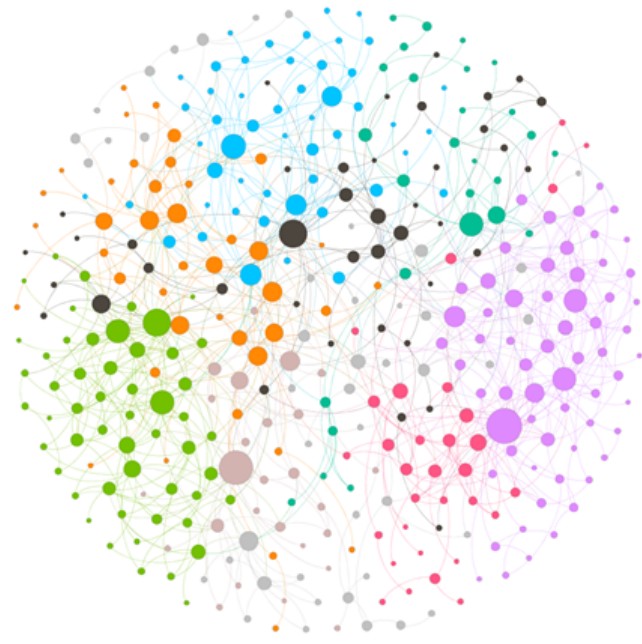

**Figure 11.** Citation Network (WOS)

**Table 5.** Citation analysis strategy.

| Database | Total Publication | Publication with Citations More Than 5 | Greatest Connected Component | |
|---|---|---|---|---|
| | | | No. of Nodes | No. of Edges |
| SCOPUS | 2601 | 1009 | 408 | 646 |
| WOS | 1858 | 494 | 375 | 886 |

The network thus created is analyzed for the measures like page rank, eigen centrality, and betweenness centrality. Tables 6 and 7 highlight the important nodes with respect to the mentioned measures. Survey papers always have a major influence on the citation network. This is underlined by the fact that 10 publications from Tables 6 and 7 are survey papers. While Liu et al. [28] reviews the general QA approaches; approaches that are based on themes like IR and deep learning are reviewed in Kolomiyets and Moens [29], Huang [30] respectively. Hoeffner et al. [23], Shah et al. [31], Athenikos and Han [32], Srba and Bielikova [33], Lopez et al. [34], Wang et al. [35], Dimitrakis et al. [36] reviews the efforts taken by the researchers in various subdomains.

**Table 6.** Citation Analysis (SCOPUS).

| Sr. No. | Reference | TOP 10 (Page Rank) | TOP 10 (Eigen Centrality) | TOP 10 (Betweenness Centrality) |
|---|---|---|---|---|
| 1. | Hirschman and Gaizauskas [37] | YES | YES | YES |
| 2. | Toba et al. [38] | YES | YES | YES |
| 3. | Lopez et al. [39] | YES | YES | YES |
| 4. | Kolomiyets and Moens [29] | YES | YES | YES |
| 5. | Zhao et al. [40] | YES | YES | YES |
| 6. | Liu et al. [28] | NO | YES | YES |
| 7. | Kwok et al. [41] | YES | NO | YES |
| 8. | Shah et al. [31] | NO | YES | YES |
| 9. | Wang et al. [11] | YES | NO | YES |
| 10. | Khodadi and Abadeh [42] | NO | YES | NO |
| 11. | Athenikos and Han [32] | NO | YES | NO |
| 12. | Nguyen et al. [43] | NO | YES | NO |
| 13. | Burke et al. [44] | YES | NO | NO |
| 14. | Soricut and Brill [45] | YES | NO | NO |
| 15. | Huang [30] | NO | NO | YES |
| 16. | Dong et al. [46] | YES | NO | NO |

**Table 7.** Citation Analysis (WOS).

| Sr. No. | Reference | TOP 10 (Page Rank) | TOP 10 (Eigen Centrality) | TOP 10 (Betweenness Centrality) |
|---|---|---|---|---|
| 1 | Srba and Bielikova [33] | YES | YES | YES |
| 2 | Lopez et al. [34] | YES | YES | YES |
| 3 | Fader et al. [47] | YES | YES | YES |
| 4 | Toba et al. [38] | NO | YES | YES |
| 5 | Kolomiyets and Moens [29] | YES | NO | YES |
| 6 | Rodrigo and PeÃ±as [48] | NO | YES | YES |
| 7 | Zou et al. [49] | YES | YES | NO |
| 8 | Hoeffner et al. [23] | YES | YES | NO |
| 9 | Wang et al. [35] | YES | YES | NO |
| 10 | Zhao et al. [40] | YES | YES | NO |

**Table 7.** *Cont.*

| Sr. No. | Reference | TOP 10 (Page Rank) | TOP 10 (Eigen Centrality) | TOP 10 (Betweenness Centrality) |
|---|---|---|---|---|
| 11 | Qiu and Huang [50] | YES | NO | YES |
| 12 | Dimitrakis et al. [36] | NO | NO | YES |
| 13 | Moldovan et al. [51] | YES | NO | NO |
| 14 | Pal et al. [52] | NO | YES | NO |
| 15 | Burke et al. [44] | NO | NO | YES |
| 16 | Figueroa and Neumann [53] | NO | NO | YES |

### 3.2.2. Co-Citation Analysis

Co-citation is a fundamental technique for identifying the various themes in the research fields. In the co-citation network, two publications are connected if cited by the third paper. Hence, publications appearing together in reference sections are the same thematically, enabling one to identify seminal references and underlying themes. The main disadvantage of Co-citation analysis is that it has a high bias toward publications with more citations. Thus, ignoring the latest or niche publications Donthu et al. [10].

Co-citation analysis is also done on both the datasets separately on cited references. The full counting method from VOSViewer has been used to generate the network map. Table 8 discusses the co-citation strategy. The threshold for considering the reference was set to a minimum of 5 citations. 407 out of 56,134 from SCOPUS and 1125 out of 27,426 from WOS references met the threshold criteria. It is difficult to visualize the network correctly with these many nodes and edges. Hence for visualization, we have shown a smaller network formed with minimum citation criteria of 25. However, for network analysis, a complete network has been considered. Smaller networks for SCOPUS and WOS are shown in Figures 12 and 13 respectively. These visualizations are created using Linlog/modularity normalization using the VOSVeiwer tool.

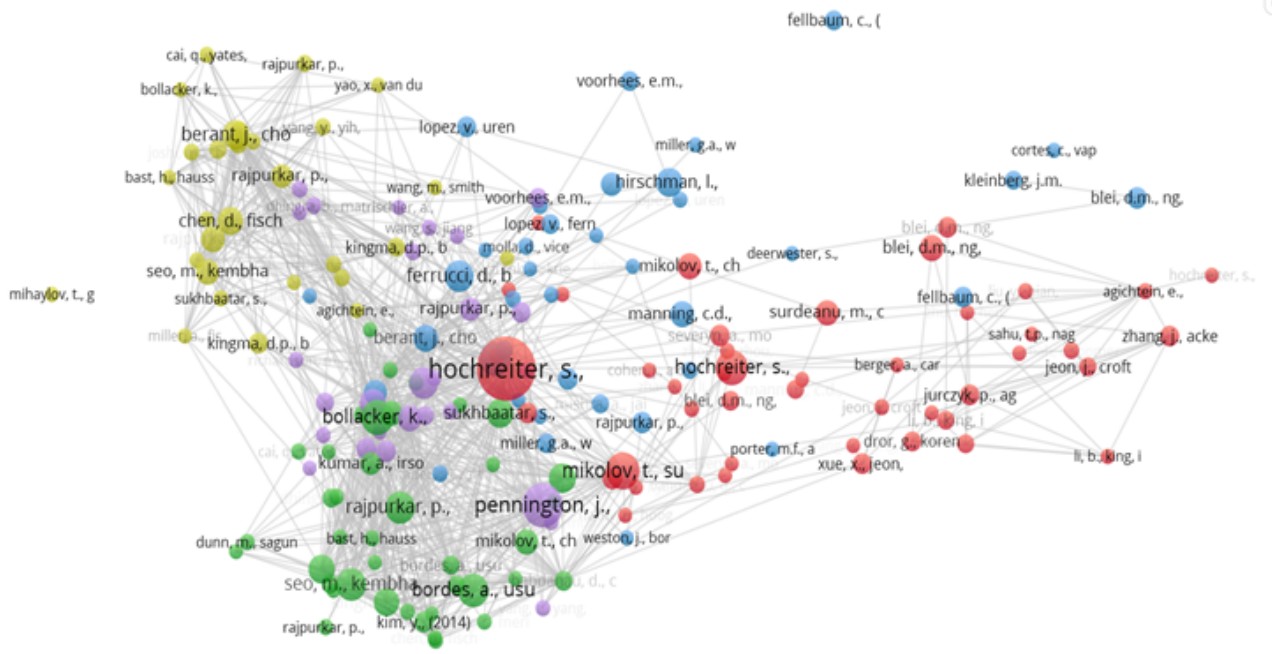

**Figure 12.** Co-citation (Scopus).

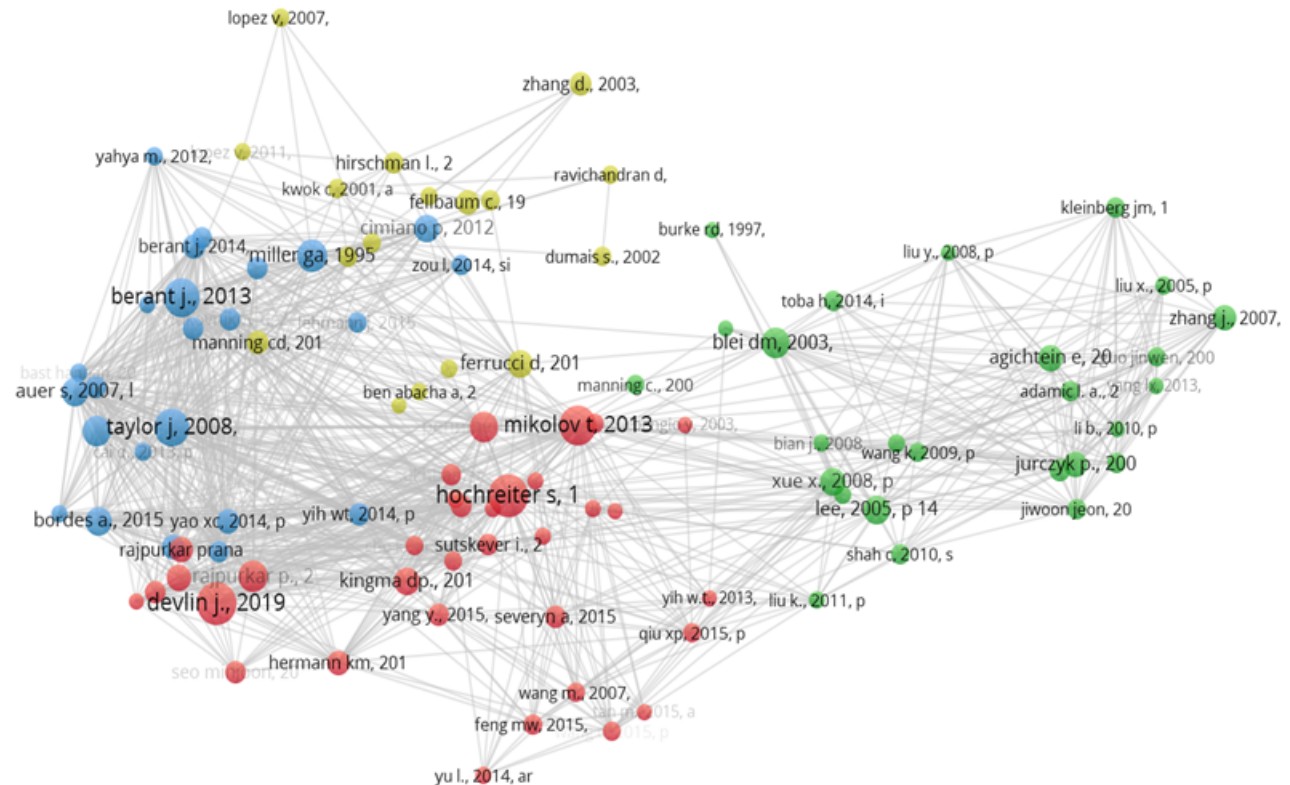

**Figure 13.** Co-citation (WOS).

**Table 8.** Co-citation analysis strategy.

| Database | Total Cited References | References with Citation More Than 5 | Greatest Connected Component | |
|---|---|---|---|---|
| | | | Number of Nodes | Number of Edges |
| SCOPUS | 56,134 | 407 | 403 | 5954 |
| WOS | 27,426 | 1125 | 1000 | 57,299 |

Page-rank, Eigen Centrality, and Betweenness centrality measures were calculated through network analysis using Gephi. Seminal references ranked by these measures are listed in Table 9 and 10. Most of the papers in these lists are highly appreciated publications and foundational papers in respective sub-themes.

**Table 9.** Co-Citation (SCOPUS).

| Sr. No. | Reference | TOP 10 (Page Rank) | TOP 10 (Eigen Centrality) | TOP 10 (Betweenness Centrality) |
|---|---|---|---|---|
| 1. | Hochreiter and Schmidhuber [54] | YES | YES | YES |
| 2. | Pennington et al. [55] | YES | YES | YES |
| 3. | Mikolov et al. [56] | YES | YES | YES |
| 4. | Bollacker et al. [57] | YES | YES | NO |
| 5. | Hermann et al. [1] | YES | YES | NO |
| 6. | Devlin et al. [4] | NO | YES | YES |
| 7. | Sutskever et al. [58] | YES | YES | NO |
| 8. | Rajpurkar et al. [5] | YES | YES | NO |
| 9. | Seo et al. [59] | YES | YES | NO |

**Table 9.** *Cont.*

| Sr. No. | Reference | TOP 10 (Page Rank) | TOP 10 (Eigen Centrality) | TOP 10 (Betweenness Centrality) |
|---------|-----------|--------------------|---------------------------|--------------------------------|
| 10. | Wang et al. [11] | YES | YES | NO |
| 11. | Ferrucci [60] | YES | NO | NO |
| 12. | Chen et al. [61] | NO | NO | YES |
| 13. | Miller [62] | NO | NO | YES |
| 14. | Radford et al. [63] | NO | NO | YES |
| 15 | Pedregosa [64] | NO | NO | YES |

**Table 10.** Co-Citation (WOS).

| Sr. No. | Reference | TOP 10 (Page Rank) | TOP 10 (Eigen Centrality) | TOP 10 (Betweenness Centrality) |
|---------|-----------|--------------------|---------------------------|--------------------------------|
| 1. | Hochreiter and Schmidhuber [54] | YES | YES | YES |
| 2. | Mikolov et al. [56] | YES | YES | YES |
| 3. | Berant et al. [65] | YES | YES | YES |
| 4. | Bollacker et al. [57] | YES | YES | YES |
| 5. | Pennington et al. [55] | YES | YES | YES |
| 6. | Devlin et al. [4] | YES | YES | YES |
| 7. | Blei et al. [66] | YES | YES | YES |
| 8. | Ferrucci [60] | YES | YES | YES |
| 9. | Miller [62] | YES | NO | YES |
| 10. | Rajpurkar et al. [5] | YES | YES | NO |
| 11. | Manning et al. [67] | NO | NO | YES |
| 12. | Lehman and Stanley [68] | NO | YES | NO |

### 3.2.3. Co-Word Analysis

While Co-citation helps to find the intellectual structure of the domain, it contributes significantly less with respect to publication content. On the other hand, keywords carry scientific concepts, ideas, and knowledge. Hence, Co-word analysis is an important technique to identify the research trends. So, it is worthwhile to look at keywords used in collected documents. Unfortunately, inconsistency creeps in the keywords in the form of synonyms, plural, use of abbreviations, or inconsistent punctuation. A few sample examples of this inconsistency are given in Table 11.

**Table 11.** Keyword Synonyms.

| Sr. No. | Keyword | Synonym |
|---------|---------|---------|
| 1. | question answer | question answering, question-answer, QA, question answering system. Question answering systems |
| 2. | natural language processing | nlp, NLP, Natural Language Processing |
| 3. | Convolution Neural Network | CNN, cnn, Convolution neural networks |

These similar keywords are clustered together by applying N-gram fingerprint key collision algorithm and a nearest neighbor algorithm. These clusters are then manually inspected, and synonyms are replaced by one consistent representation. 3697 unique keywords were identified from the 'merged-dataset'. The top 10 keywords are given in

Figure 14. As expected, Question-answering and Natural Language Processing are the two most used keywords by authors. Few studies have also used Information Retrieval, deep learning, Machine learning, Knowledge-base, community question answering, Knowledge graph as keywords.

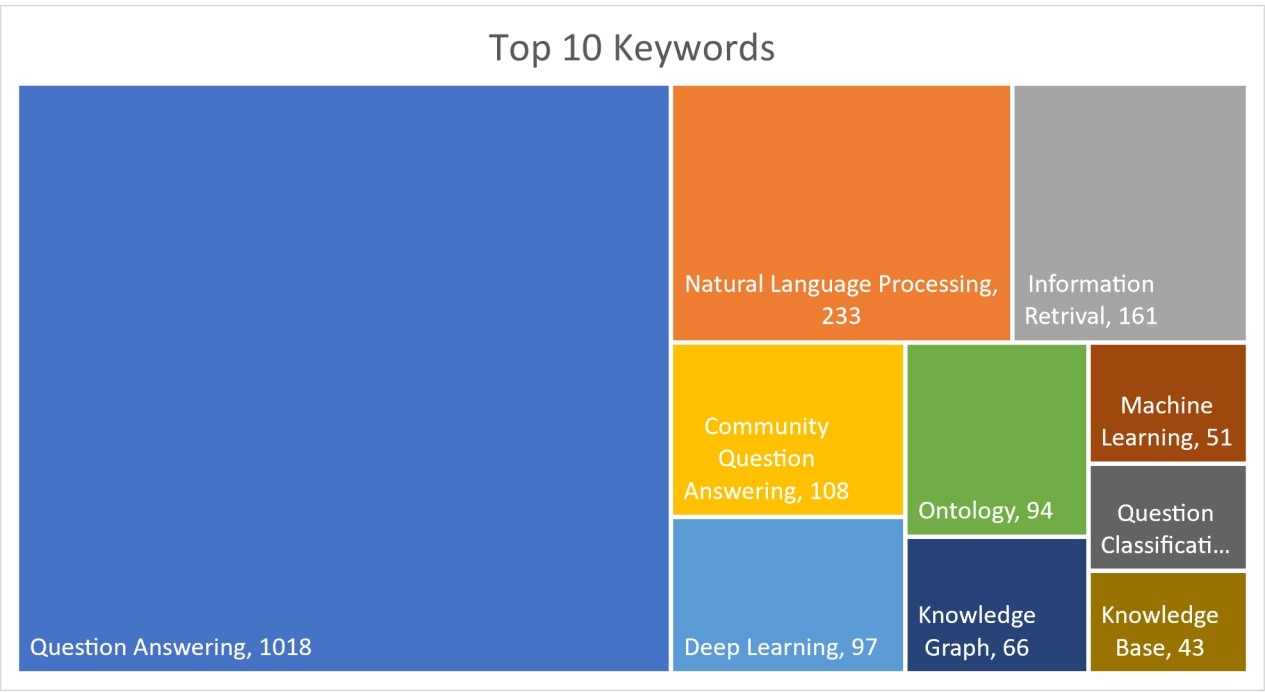

**Figure 14.** Top 10 Keywords.

As discussed in Section 3.1.1, research in the QA domain gathered momentum in the 2000s, and the last 5–6 years are critically important. To further understand the undercurrent of the research trend, this duration is split into time frame 1: 2001–2016 and time frame 2: 2017–2021 for further analysis. Table 12 shows the keywords used in those time frames.

**Table 12.** Keyword trend.

| 2017–2022 | | 2001–2016 | |
|---|---|---|---|
| **Author Keywords** | **Count** | **Author Keywords** | **Count** |
| question answering | 472 | Question Answering | 536 |
| natural language processing | 140 | Information Retrieval | 95 |
| deep learning | 93 | Natural Language Processing | 93 |
| information retrieval | 65 | Ontology | 70 |
| knowledge graph | 64 | Community Question Answering | 50 |
| community question answering | 58 | Passage Retrieval | 31 |
| knowledge base | 35 | Machine Learning | 28 |
| question classification | 29 | Semantic web | 25 |
| convolution neural network | 25 | Information extraction | 25 |
| Ontology | 24 | Query expansion | 24 |

If we ignore the generic QA and NLP keywords, other important keywords from timeframe-1 are Information Retrieval, Ontology, Passage Retrieval, and Semantic web. At the same time, those from timeframe-2 are Deep Learning, Knowledge graph, Convolution Neural Network. This usage of keywords shows that efforts are now shifted to neural

network-based and knowledge graph-based solutions. Same is also evident from Figure 15. Yellow cluster in figure represents the recent trends in keyword usage.

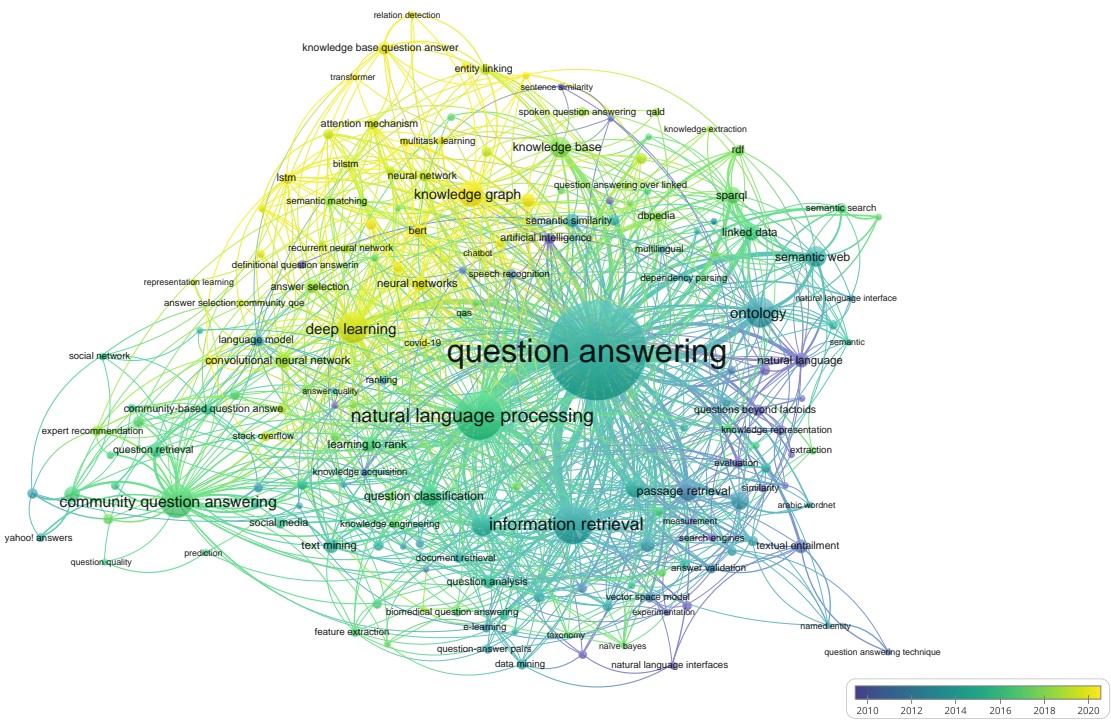

**Figure 15.** Co-occurrence Overlay Visualization.

Co-occurrence analysis is performed on author keywords using VOSViewer. Keywords occurring with more than five are only considered for the analysis. 178 keywords out of 3697 met the criteria. Top-15 keywords are listed in Table 13 for the question-answering systems.

**Table 13.** Top 15 keyword.

| Keyword | Links | Occurrences | TLS |
|---|---|---|---|
| question answering | 155 | 1018 | 1302 |
| natural language processing | 96 | 233 | 437 |
| information retrieval | 75 | 161 | 326 |
| deep learning | 63 | 97 | 184 |
| Ontology | 47 | 94 | 169 |
| community question answering | 54 | 108 | 142 |
| semantic web | 27 | 42 | 122 |
| machine learning | 43 | 51 | 109 |
| knowledge graph | 42 | 66 | 97 |
| information extraction | 29 | 39 | 81 |
| knowledge base | 31 | 43 | 75 |
| Sparql | 27 | 26 | 74 |
| question classification | 36 | 45 | 74 |
| natural language | 27 | 22 | 71 |
| passage retrieval | 26 | 40 | 69 |

Figure 16 is the density map of the Co-Occurrence graph and illustrates the crucial aspects of the QA systems. It shows four prominent clusters and two small clusters. The first cluster offers the generalized view of the QA system. The second cluster includes the solution involving the Deep Learning-based solutions, the latest SOTA model-based attention mechanism, or RNNs. The third cluster talks about a very important subset of the QA system, i.e., Community Question Answer (CQA) systems. The fourth cluster is about ontology and semantic-related works. The fifth and Sixth clusters are less significant ones. Table 14 lists the top 10 keywords from each cluster.

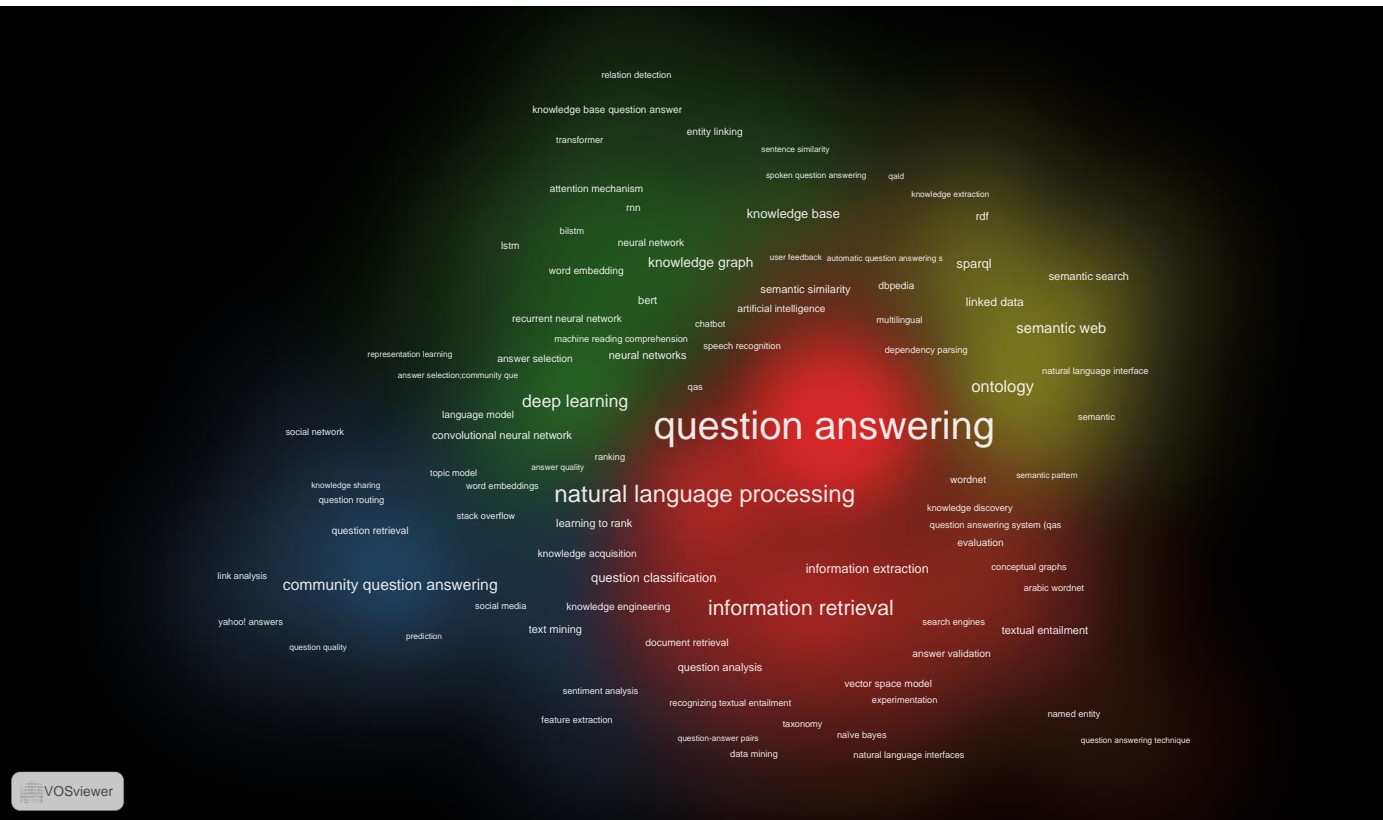

**Figure 16.** Co-occurrence cluster.

**Table 14.** Cluster-wise top keywords.

| Keyword | Cluster | TLS | Keyword | Cluster | TLS |
|---|---|---|---|---|---|
| question answering | 1 | 1302 | ontology | 4 | 169 |
| natural language processing | 1 | 437 | semantic web | 4 | 112 |
| information retrieval | 1 | 326 | sparql | 4 | 74 |
| machine learning | 1 | 109 | natural language | 4 | 71 |
| information extraction | 1 | 81 | linked data | 4 | 64 |
| question classification | 1 | 74 | rdf | 4 | 35 |
| passage retrieval | 1 | 69 | semantic search | 4 | 33 |
| query expansion | 1 | 65 | dbpedia | 4 | 30 |
| answer extraction | 1 | 45 | artificial intelligence | 4 | 23 |
| question analysis | 1 | 42 | semantics | 4 | 21 |

**Table 14.** *Cont.*

| Keyword | Cluster | TLS | Keyword | Cluster | TLS |
|---|---|---|---|---|---|
| deep learning | 2 | 184 | knowledge engineering | 5 | 22 |
| knowledge graph | 2 | 97 | knowledge acquisition | 5 | 20 |
| knowledge base | 2 | 75 | big data | 5 | 15 |
| convolutional neural network | 2 | 45 | summarization | 5 | 14 |
| neural networks | 2 | 40 | non-factoid question answering | 5 | 13 |
| bert | 2 | 33 | user interaction | 5 | 9 |
| lstm | 2 | 29 | information seeking | 5 | 4 |
| neural network | 2 | 29 | social question answering | 5 | 3 |
| answer selection | 2 | 28 | data mining | 6 | 15 |
| attention mechanism | 2 | 26 | cross-lingual question answering | 6 | 12 |
| community question answering | 3 | 142 | natural language interfaces | 6 | 10 |
| text mining | 3 | 46 | | | |
| learning to rank | 3 | 42 | | | |
| expert finding | 3 | 32 | | | |
| question retrieval | 3 | 30 | | | |
| language model | 3 | 21 | | | |
| tf-idf | 3 | 19 | | | |
| question routing | 3 | 18 | | | |
| crowdsourcing | 3 | 17 | | | |
| expert recommendation | 3 | 17 | | | |

## 4. Qualitative Analysis

This section focuses on the qualitative survey of the QA domain. Table 12 shows the recent shift in approaches of QA systems. Since WWW has matured and now stores the information more structured way, researchers are more inclined toward Knowledge Base/Graph-based QA systems. Same is also evident from Section 3.1.2. Hence, we have considered the systems from this sub-domain for qualitative analysis. This section is organized as Section 4.1 paints the general picture of the QA domain, while Section 4.2 focuses on KB-based QA systems (KBQA). Graph Neural Network (GNN) has shown better results recently in processing the data represented in the form of graphs. Hence, Section 4.3 explores the GNN-based solutions for KBQA.

### 4.1. General Approaches

One of the first well-known efforts in the QA domain was BASEBALL, developed by Jr. Green B.F. and Laughery [69]. QA systems have been changed drastically since then. We can broadly categorize the attempts into three important classes: linguistic approach, statistical approach, and Pattern Matching approach. The linguistic approach attempts to understand the natural language text by deploying various techniques like tokenization, POS tagging, and parsing. Few notable efforts includes Jr. Green B.F. and Laughery [69], M. et al. [70], Clark et al. [71], Mishra et al. [72], Bobrow et al. [73], Xiaoyan et al. [74]. Due to the availability of huge information, a statistical method for QAS has increased recently. Kim et al. [75], Mansouri et al. [76], Liu and Peng [77], Moschitti [78], Zhang and Zhao [79], Quarteroni and Manandhar [80] uses the SVM statistical method for either question classification or identifying the feature of words or as a text classifier. Another statistical method, N-gram mining is used in Soricut and Brill [81], Berger et al. [82], to form a chunk from a question. Cai et al. [83] uses Sentence Similarity Model for Web-based Chinese QA system

with answer validation; while maximum entropy model for question/answer classification is used in Ittycheriah et al. [84] along with various N-gram or bag of words features.

Surface-based pattern matching technique is mostly used for factoid questions, while closed domain questions are answered using template-based pattern matching techniques. Molla and Vicedo [85], Ravichandran and Hovy [86], Cui et al. [87], Du et al. [88] are a few surface-based pattern matching techniques where patterns are either handcrafted or automatically detected. Template-based techniques are also widely used e.g., RDF in Unger et al. [89], Zhang and Zou [90], To and Reformat [91], FAQ in Burke et al. [44], Liu et al. [92], Otsuka et al. [93], and SPARQL Cocco et al. [94], Hu et al. [95]. Other than these, few hybrid approaches try to combine more than one technique. Kwok et al. [96] is an example of integration of linguistic techniques and pattern matching, and Wang et al. [3] uses a rule-based approach and SVM; whereas surface pattern matching and entropy are combinedly used in Xia et al. [97].

### 4.2. Knowledge Base-Based Approaches

Diefenbach et al. [98] divides QA process into 5 tasks: question analysis, phrase mapping, disambiguation, query construction, and querying distributed knowledge. Question analysis deals with the syntactic features and extracts the information about the question. In phrase mapping entity identified in the question analysis is mapped to the corresponding highest probability resource from the KG. With disambiguation right resources are selected for the entities in phrase mapping. Query construction deals with constructing a SPARQL query that can be used to dig the information from KG. Sometimes we may have to retrieve the information from more than one KB; hence distributed knowledge task comprises the techniques related to that. Critical phases/sub-tasks in each task are explained in Figure 17. Not all QA systems have this clear classification of tasks; however, every QA system must perform these tasks.

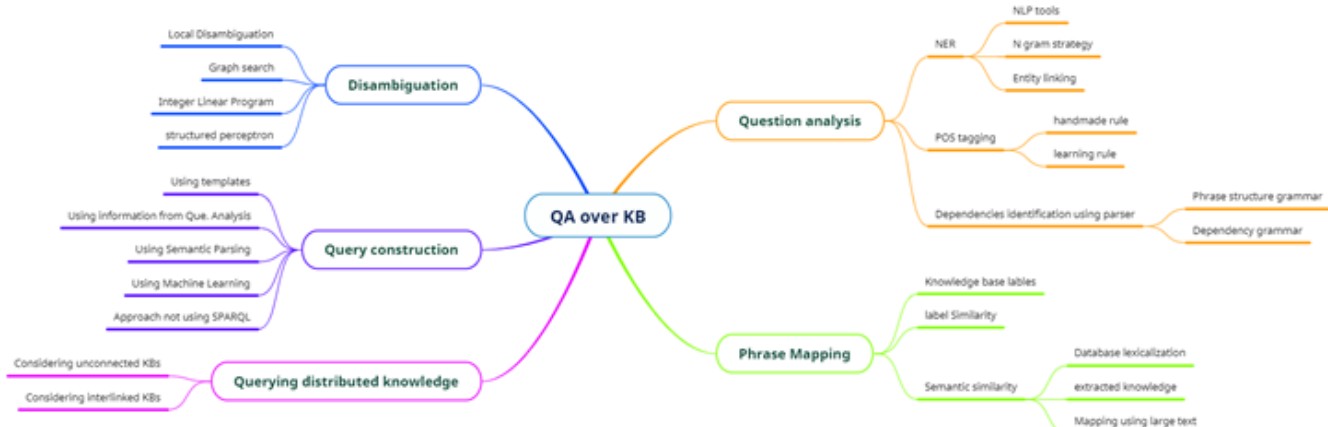

**Figure 17.** Important Task in KBQA

### 4.2.1. Initial Data Transformations

A crucial part of any QA system is understanding the question and its context. Hence, many KBQA performs the question analysis by adopting the dependency tree or Neural Network (NN)-based classification approach. In the dependency tree approach, key information about the question is extracted from the dependency tree generated. Many KBQA systems like Nicula et al. [99], Le et al. [100], Shin et al. [101] use standard dependency tree generation methods. Phase structure grammar [102] and feature-based grammar [103] have been used to generate the dependency tree, but the most prominent method uses the parsing tools like TALN [104] and Stanford Parser [105–107]. Hu et al. [95] proposed the Valuable Dependency Parser, which uses the Stanford Parser for initial parsing, and then a few tags are prioritized for query generation. While the dependency parser gives the relation between the words, constituency parsing uses context-free grammar and divides the statements into sub phrases. Zhu et al. [108] generates constituency tree, which is

then matched to KB using graph traversal technique. NN-based methods, [109–111] are also used for question or entity classification. But due to the existence of many types of question/entity, these methods perform sub-optimally.

Entity linking is another critical task in a QA system. It consists of two subtasks, viz, Named Entity Recognition (NER) and Disambiguation of extracted entities to correct entities in KB. Apart from classical NER methods, Probabilistic Graphical methods like the Maximum Entropy Markov Model (MEMM) and Conditional Random Field (CRF) are also very popular as NER. Chen et al. [112] uses two-stage MEMM as NER, whereas Bach et al. [113], Hu et al. [114], Wu et al. [115], Sui [116] uses the CRF method and various RNN models.Hu et al. [114] uses BiLSTM to capture the context while the CRF layer generates probability distribution for tag sequence. Bach et al. [113] proposes the three-stage NER model. The first stage is the CNN model that captures the word level representations, while BiLSTM in the second stage captures the sentence level representations. These representations are then given to the third stage, a CRF-based inference layer for named entity detection. However, the BERT-BiLSTM-CRF-based NER method, proposed in Sui [116] has state-of-the-art results.

To map the extracted entities to entities in KB, many QA systems have used the existing tools like Dbpedia spotlight [106,109,110,117–120], S-MART [114]. The similarity measures like Jaro-Vinkler [101], UMBC+LSA [104], and Siamese LSTM [121] have also been used for mapping the entities. Few QA systems have also used approaches based on NN [122], Hierarchical RNN [16], BiLSTM [123], and BERT [124,125].

### 4.2.2. Architectural Classification

Almost all the QA systems can be classified in four different architectures, viz., Semantic Parsing-based (Figure 18), Subgraph Matching-based (Figure 19), Template-Based (Figure 20), and Information Extraction (IE)-Based (Figure 21). These architectures are discussed in the following section one by one.

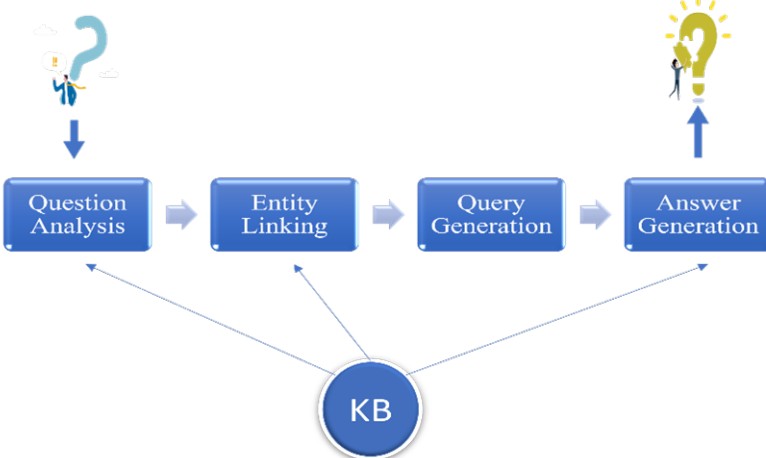

**Figure 18.** Semantic Parsing Architecture.

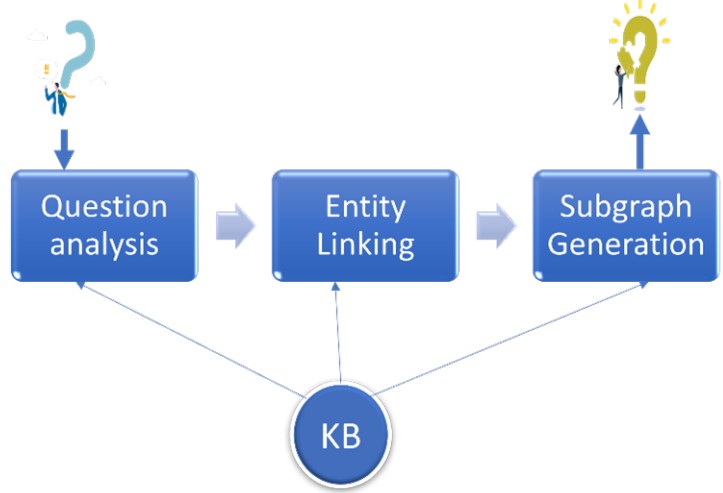

**Figure 19.** Subgraph-Based Architecture.

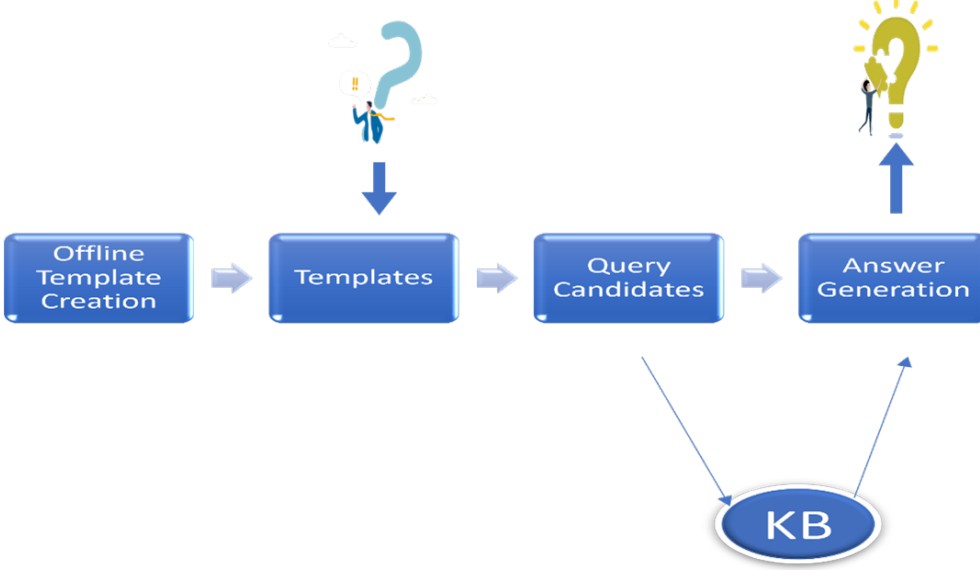

**Figure 20.** Template-Based Architecture.

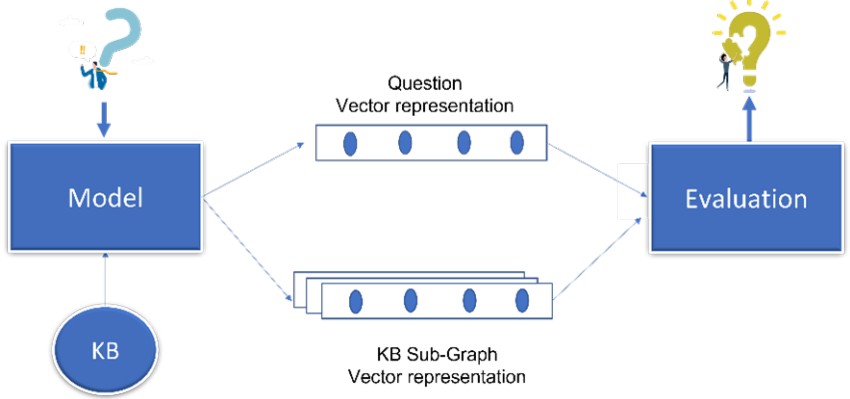

**Figure 21.** Information Extraction-Based Architecture.

- Semantic Parsing-Based Methods:

A semantic parsing-based method maps natural language question to logical form query, which can then be executed against the knowledge base to retrieve the answer. Dependency tree and question analysis done in initial data transformation help generate the query. Semankey [126] generates the multiple trees for each combination of extracted entities by considering the underlying ontology and using BFS-based algorithm. Various trees are generated by starting the BFS from the main entity of interest. Each tree is then converted into a SPARQL query and executed against the KB. Instead of generating multiple trees, method proposed by Maheshwari et al. [127] generates multiple candidate paths starting from the main entity mapped in KB. These paths, along with the question, are encoded using BiLSTM and ranked by calculating the dot product of the question encoding vector and path encoding vector. Alternatively, in the method proposed in Zafar et al. [128], latent representations of candidate path and questions are obtained using Tree-LSTM, and the similarity function is used to rank these representations.

Generator-Reranker architecture is proposed by Inan et al. [129]. The generator produces a list of potential candidates, and the reranker ranks these candidates based on the similarity between each candidate and the input sentence.Lu et al. [130] have proposed the recall-oriented information extraction method. Query generation is then viewed as a linear programming problem and solved using the Steiner tree. There are many other approaches for query generation like the use of DAG [131], Hidden Markov Model [132], Siamese Neural Network model [109], and RNN [133].

The semantic pipeline for KBQA seems natural and intuitive; however, looking at the current research and publication trends, we can conclude that this type of solution is already maturing, and other types of solutions should be explored for future research.

- Sub-graph-based method:

Logical form query execution is like finding the subgraph. Hence, the subgraph matching-based method performs the initial data transformation similar to the semantic parsing method. However, instead of generating the formal query, it compiles the query graph and then models it as a subgraph finding problem.

Majority of the approaches [105,106,119] extracts several triplet pattern (Es,R,Ed). Es and Ed are the source and destination entities, and R is their relation. These triplets are identified using initial transformations discussed earlier. Jin et al. [105] then retrieves the candidate subgraph from KB and uses semantic similarity to evaluate the matching between triplets and retrieved sub-graph. Bakhshi et al. [119] proposes the novel data-driven graph similarity framework, where disambiguation is handled with the help of user input. Then the QA problem is remodeled as Integer Linear Programming (ILP) and solved using the proposed golden graph alignment method and standard optimizers. On the other hand, method in Li et al. [106] uses extracted triplets to build the query graph and resolves the ambiguities by finding isomorphic graphs of this query graph from KB. A semantic similarity between a semantic vector of edge in query graph and path in KB is found.

Hu et al. [114] uses a rule-based approach to generate a semantic query graph (SQG) using the state-transition paradigm. First, entities are recognized using BiLSTM and CRF-based model. Then, these recognized entities undergo four state transition operations: Connect, Merge, Expand, and Fold to generate SQG. While method in Hu et al. [114] generates SQG, Hu et al. [107] generates the dependency tree, and Zhu et al. [108] generates the constituency tree. Such generated tree/SQG is then found in KB using graph traversal technique and path ranking method.

- Template-based method:

In template-based methods, initial data transformation leads to a template for the question, which is then matched with templates stored in a repository. These templates are nothing but pseudo-query with some slots. The best matching template is selected and instantiated with statistical methods to retrieve the SPARQL query [89]. Generating the

template repository and extracting the query structure from the question are two essential tasks in template-based systems.

Ever since this type of solution was proposed in Unger et al. [89], researchers have come up with many template-based QA systems. Most of the approaches use handcrafted rules to extract the query structure of question [120,134] and hand-crafted templates [115,134].

Vollmers [135] proposes TeBaQA, an isomorphic graph-based approach. Basic graph patterns of some SPARQL queries are maintained in the repository. Using ML classifier, natural language questions are then classified into these isomorphic basic graph patterns (i.e., templates). To select the appropriate template from a database, Wu et al. [115] identifies the question type using LSTM and uses the BiLSTM+CRF model as a NER tool to extract the entities. These entities are used to instantiate the template. Apart from Neural network methods, few solutions like Abujabal et al. [18], To and Reformat [91] involve the dependency tree generated from the question. The isomorphic tree of this dependency tree is selected from the template repository, and the SPARQL query is generated.

Though this solution offers better results, we have found very few systems. One of the possible reasons is that the system is too dependent on the collection of templates. At the same time, researchers are also moving toward a more end-to-end approach envolving newer deep learning and transfer learning solutions, discussed in the next section.

- • Information Extraction-based methods:

In IE-based architecture, question and KB sub-graph are encoded into common embedding space using some machine learning approach and directly matched them using a predefined scoring function. To capture the more accurate feature for encoding, initial data transformations are done sometimes.

As discussed in earlier sections, there have been many ML solutions for initial transformation tasks. Similarly, there are many end-to-end solutions for QA tasks. These approaches usually do not need hand-crafted features or rules designed by experts and can scale better to large and complex KBs. Two main DNN architectures that are mainly used are CNN and RNN. Dai et al. [136], Yin et al. [137] proposes an End-to-end neural network model for answering factoid questions, whereas Golub and He [138] uses A character-level encoder-decoder framework- Seq2Seq LSTM model for the same task. Lukovnikov et al. [12] also uses a neural network for answering simple questions end-to-end, leaving all decisions to the model. Wang et al. [139,140], Xie et al. [141] are a few methods that use CNN to get the embedding/vector representations.

However, recently researchers have started using hybrid ML solutions where individual subtasks are solved using an appropriate ML model. Budiharto et al. [142] uses two approaches, RNN-based and CNN-based encoder. The output of these encoders forms the hidden vector. Bidirectional Attention Flow (BiDAF) is then used to match the hidden vector of KB and the hidden vector of the question. Song et al. [143] proposed a method in whitch Semantic features are learned using LSTM. The attention mechanism is added to focus on a particular part of the answer to generate the embeddings. Qu et al. [144] trains three separate models. An Entity Alignment Model is a simple NN model; the Object Answering Model is SVM, while the Prediction-Verification Model is an encoder-decoder model. In Luo et al. [145], the query structure is encoded into a uniform vector representation to capture the interactions between individual semantic components of the question.

Instead of finding the vector representations directly, if we do the initial transformation and get a better understanding of a question, we can get a better vector representation. With this aim, Tong et al. [146] generates the dependency tree and proposes a Tree-structured LSTM model that accepts tree-structured input to get the embeddings.

Due to limited data availability, these LSTM-based approaches tend to overfit. Hence use of Bidirectional Encoder Representations from Transformers (BERT) is explored in Luo et al. [124], Lukovnikov et al. [147], Panchbhai et al. [148]. One can also use five BERT models for five different tasks: question expected answer type (Q-EAT), answer type (AT)

classification models, and Question Answering Model, as suggested by Day and Kuo [149]. Each BERT model is fined tuned for the respective task.

Thus, models generating vector representations avoid semantic pipelines. Due to the availability of many DNN approaches and the possibilities of combining them in various ways, many approaches are reported in the literature. LSTM-based approaches combined with attention mechanisms have reported promising results and can be further explored for betterment. On the other hand, pre-trained model-based approaches are still nascent.

*4.3. GNN-Based Approaches*

Previous efforts for QA systems in multi-hop settings were based on RNN. But DAGs used in RNNs cannot represent the rich inference. Cao et al. [150], Song et al. [151] are the first few attempts to use GNN for the same. In Cao et al. [150], entities mentioned in multiple documents are identified as nodes, and the edges represent the relations between the entities (either with-in or cross-document). The graph thus formed is then processed using GCNs to generate the multi-hop reasoning.

Instead of using a single co-reference edge, Song et al. [151] proposes to use 2 additional types of edges: 'same' and 'window.' The 'same' edge helps to connect the multiple mentions of an entity appearing far apart. This helps capture global information. The 'window' edge connects to entities mentioned in a fixed window. GCN and Graph recurrent network is then used to demonstrate the usefulness of the graph thus formed.

The bi-directional Attention Entity Graph Convolutional Network (BAG) method for multi-hop QA, proposed in Cao et al. [152], generates the graph by extracting the entities and relationships from multiple documents. Relation-aware representations for the identified entity nodes are then acquired from the Relational Graph Convolutional network (R-GCN). BAG then uses, Bi-Directional Attention mechanism to generate a more meaningful relationship between query and graph.

Tu et al. [153] introduces a Heterogeneous Document-Entity (HDE) graph to represent the knowledge obtained from multiple documents. Instead of using a node of a single type (as in all other KG approaches), heterogeneous graphs contain nodes of various types like candidate, entity, and document nodes. These different nodes relate to each other with different types of edges, representing the more structural representation. To demonstrate the effectiveness of HDE, the authors have used Graph neural network (GNN) with a message-passing algorithm.

Xiao [154] argues that instead of using the same static graph for every query, one should use a dynamic graph built explicitly for a given query. It also proposes a novel method, Dynamically Fused Graph Network (DFGN), for the same reason. The graph generation process in DFGN starts with the main entity of the given query, and then the graph is constructed by connecting the related entities around this start entity. This process continues till a probable answer is discovered.

Three main challenges of comprehending the text for QA are Reasoning ability, Explainability, and Scalability Ding et al. [7]. To tackle these issues, a method mimicking the cognitive process of humans is proposed in Ding et al. [7]. The proposed method, CogQA has two modules to build the cognitive graph. The first module is implicit extraction which extracts the meaningful entities of questions and probable answers and forms the graph of working memory. The second module is used for explicit reasoning. This module performs reasoning over the constructed graph and gathers the important hints for module one. These tips are further used by module one in the extraction of next hop entities. This iterative process continues until all probable answers are found. The final answer will depend on the second module's reasoning in the final round. They have proposed the BERT-based extraction module and GNN-based reasoning module.

Vakelenko et al. [155] proposes another approach for the complex QA, QAmp. Here, important entities and relations are identified and mapped to the relevant portion of a graph. This approach uses unsupervised message passing to propagate the confidence

values across the graph. These scores lead to the correct answer entities. At last, these scores are aggregated depending on the type of question.

Due to the natural language's flexibility, the questions are highly unstructured, whereas knowledge graphs maintain the information in a very structured way. So, a way to convert an unstructured sentence to a structured query to process the information in KG is needed. Also, ambiguation in the question itself leads to wrong answers. Many disambiguation models like Hu et al. [95], Xiong et al. [156], Zheng et al. [157], Zhu and Iglesias [158] has been proposed. Disambiguation can be reduced by asking a few questions to the asker; hence Zheng et al. [157] proposes a novel interactive method that allows users to use natural language to query the KG. While Xiong et al. [156], Zhu and Iglesias [158] use semantic tools for disambiguation, the NLQSK framework is proposed in Hu et al. [95], where top k-SPARQL statements are generated for given questions and keyword search is used to fetch the neighboring information for unmapped entities and finally these two are combined to produce the answer.

## 5. Summary

To explore the complete domain space of QAS, publications are extracted from WOS and SCOPUS databases. Both sets of extracted publications are then merged together by considering the MCA and overall 4459 publications are retained. Further, inconsistencies are removed by using 'n-gram fingerprint key collision' algorithm. Performance analysis is done on this merged data-set however, science mapping is done individually on WOS and SCOPUS data. The answers to the research questions can be found in the follwing important findings.

1. Even though publications in the QAS domain are from the 1960s, 50% are from the last six years, indicating that QA is attracting many researchers nowadays.
2. Penas A, Nakov P, Moschitti A, Lehman J are influential contributors with more than 15 publications. China is the most productive country and, along with USA, India, and Germany, contributes to more than 50% of the overall publications.
3. We have also identified the highly cited publications in the last 5 years from both SCOPUS and WOS (listed in Tables 3 and 4).
4. Using network analysis measures like Page rank, Eigen centrality, and Betweenness centrality on the citation graph, we have identified the most influential publications (listed in Tables 6 and 7).
5. Co-citation analysis highlighted the QA domain's seminal and foundational publications (listed in Tables 9 and 10).
6. Co-occurrence analysis helped us to identify the four major sub-domains of QAS. This analysis also helped us conclude that neural network and knowledge base-based solutions are recent research trends.
7. This study also gave a summary of important methods in KB-based solutions. We classified the approaches into four important classes and discussed the various approaches to perform the sub-tasks in each class.
8. Finally, we have also discussed the approaches belonging to one of the most upcoming and promising areas, i.e., GNN-based approaches.

## 6. Conclusions

Bibliometric analysis of QAS has been done along with a literature review of two important subdomains. This study highlights the important constituents of QAS domain along with seminal and foundational publications. We have also observed that neural network and knowledge graph-based solution is the current research trend.

Out of four architecture types of KB-based solutions, semantic parsing-based approach is already maturing and provides less scope for future novel work. Subgraph-based and template-based approaches shows promising results, but due to heavy dependence on hand-crafted features, they are not explored completely. However, Information Extraction-

based approach provides end-to-end solution for all sub-tasks and hence most of the research is moving in this direction.

**Author Contributions:** Conceptualization, B.Z. and S.M.; Formal analysis, B.Z. and R.V.B.; Investigation, B.Z. and R.V.B.; Methodology, B.Z. and S.M.; Project administration, S.M., K.S., D.R.V. and K.K.; Software, B.Z., S.M. and R.V.B.; Supervision, S.M., K.S., D.R.V. and K.K.; Validation, B.Z., S.M., K.S., D.R.V., K.K. and R.V.B.; Visualization, B.Z. and R.V.B.; Writing—original draft, B.Z.; Writing—review & editing, S.M., K.S., D.V., K.K. and R.V.B. All authors have read and agreed to the published version of the manuscript.

**Funding:** This research received no external funding.

**Institutional Review Board Statement:** Not applicable.

**Informed Consent Statement:** Not applicable.

**Data Availability Statement:** Not applicable.

**Conflicts of Interest:** The authors declare no conflict of interest.

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
