# Peer review of "Question Answer System: A State-of-Art Representation of Quantitative and Qualitative Analysis"

_2504-2289, doi:10.3390/bdcc6040109_

Round 1

Reviewer 1 Report

Interesting paper and a lot of information. 

The manuscript is clear, relevant for the field and presented in a
well-structured manner, and I personally like the structure.

Using the "network graph" gives the paper an edge and an additional scientific contribution.

Try to add the purpose of choosing these methods for this research and re-read to address the minor grammatical errors. 

Author Response

Response to Reviewer 1 Comments

Dear Reviewer,

We are grateful for your time and effort in offering your insightful comments on my manuscript. To address the majority of your recommendations, we were able to make changes. Same has been highlighted in manuscript as well. However, here is a point-by-point response to your comments and concerns.

  1. Interesting paper and a lot of information. 

NA

  1. The manuscript is clear, relevant for the field and presented in a well-structured manner, and I personally like the structure.

NA

  1. Using the "network graph" gives the paper an edge and an additional scientific contribution.

NA

  1. Try to add the purpose of choosing these methods for this research and re-read to address the minor grammatical errors. 

Text is added to highlight the purpose. (see the change marked as c3) also identified grammatical errors are removed.

Reviewer 2 Report

Overall

This paper describes a sophisticated and comprehensive review of the state of the art of Question Answer Systems. The literature review provides a go-to reference that researchers in this domain will greatly appreciate. The analyses of the literature are wide ranging and the results are presented in an easy-to-comprehend manner with clear concise textual descriptions supported by visuals. The visuals are well-formatted and suited to the purpose.

Despite the many good points in this paper, there are a number of issues that need to be resolved prior to receiving a recommendation to publish. Most of the suggestions relate to the organization and framing of the paper rather than to the literature review and analyses per se. As such, the issues should be relatively easy to address.

Issues

1.  Title

There is a disjuncture between the title and the abstract/article. This paper is essentially an analysis of the extant literature on QASs, and so the title should show that. Currently, readers will not expect this to be a paper that analyzes the research literature.

2.  Abstract / Introduction 

The need for quantitative and qualitative analyses of the body of literature needs to be shown. The analyses can then be presented as solving a problem or solving multiple problems.

3.  Figure 3 

Providing an overview of the paper is laudable, but this should be in textual form and not a figure. Delete the figure and rewrite the content as a paragraph.

4.  Structure

I found it difficult to follow the argument in your paper. This could be improved by adding a section that introduces the method including the different types of analyses to be used. The rationale for each of the different types of analyses could be included there. Subsequent sections can then introduce the individual analyses.

5. Conclusion

Provide a more detailed summary showing how each of the analyses solved a particular problem or addresses a specific question.

6.  Tables 5, 8, and 14

Reformat so the tables are in the printable area.

7.  Line 283 

Insert space before section number.

Author Response

Response to Reviewer 2 Comments

Dear Reviewer,

We are grateful for your time and effort in offering your insightful comments on my manuscript. To address the majority of your recommendations, we were able to make changes. Same has been highlighted in manuscript as well. However, here is a point-by-point response to your comments and concerns.

Issues

  1. Title: There is a disjuncture between the title and the abstract/article. This paper is essentially an analysis of the extant literature on QASs, and so the title should show that. Currently, readers will not expect this to be a paper that analyzes the research literature.

We appreciate the reviewer’s insightful suggestion and agree that it would be useful to add word ‘bibliometric’ in title for more clarity. However, the ‘quantitative’ part in the title refers to bibliometric study and this paper contains both type of analysis (bibliometric and traditional literature survey). Words ‘quantitative’ and ‘qualitative’ are retained in title for alliteration purpose and use of word ‘bibliometric’ will break the rhythm.

  1. Abstract / Introduction: The need for quantitative and qualitative analyses of the body of literature needs to be shown. The analyses can then be presented as solving a problem or solving multiple problems.

Paragraph starting on line 52 is modified a bit to explains the research gap while subsequent paragraph also mentions research questions considered in study.

  1. Figure 3: Providing an overview of the paper is laudable, but this should be in textual form and not a figure. Delete the figure and rewrite the content as a paragraph.

We have revised the text to address your concerns and hope that it is now clearer. Figure 3 is now deleted and same is explained in paragraph. (See change C2)

  1. Structure: I found it difficult to follow the argument in your paper. This could be improved by adding a section that introduces the method including the different types of analyses to be used. The rationale for each of the different types of analyses could be included there. Subsequent sections can then introduce the individual analyses.

Thank you! We found your comments extremely helpful. We have added a text to provide a rationale and to introduce all the types of analysis (see change c3). Moreover, as per your suggestions, they are explained in-detail in subsequent sections. 

  1. Conclusion: Provide a more detailed summary showing how each of the analyses solved a particular problem or addresses a specific question.

Separate summary section is present for the same.

  1. Tables 5, 8, and 14: Reformat so the tables are in the printable area.

We’ve corrected the error. We apologize for our error

  1. Line 283: Insert space before section number.

This was an oversight. We’ve corrected the error. We apologize for our error

Round 2

Reviewer 2 Report

The authors have addressed all the concerns raised in the first round of reviews; and, as such, I have no more objections to publication. I look forward to seeing this article in press.